# Advancing Interpretability of CLIP Representations with Concept Surrogate Model

**Nhat Hoang-Xuan**
University of Florida

**Xiyuan Wei**
Texas A&M University

**Wanli Xing**
University of Florida

**Tianbao Yang**
Texas A&M University

**My T. Thai**[*]
University of Florida

## Abstract

Contrastive Language-Image Pre-training (CLIP) generates versatile multimodal embeddings for diverse applications, yet the specific information captured within these representations is not fully understood. Current explainability techniques often target specific tasks, overlooking the rich, general semantics inherent in the representations. Our objective is to reveal the concepts encoded in CLIP embeddings by learning a surrogate representation, which is expressed as a linear combination of human-understandable concepts evident in the image. Our method, which we term EXPLAIN-R, introduces a novel approach that leverages CLIP's learned instance-instance similarity to train a surrogate model that faithfully mimics CLIP's behavior. From the trained surrogate, we derive concept scores for each input image; these scores quantify the contribution of each concept and act as the explanation for the representation. Quantitative evaluations on multiple datasets demonstrate our method's superior faithfulness over the baseline. Moreover, a user study confirms that our explanations are perceived as more relevant, complete, and useful. Our work provides a novel approach for interpreting CLIP image representations, enhancing the user interpretability of representations and fostering more trustworthy AI systems.

## 1 Introduction

The CLIP model [1] exemplifies the success of representation learning, which aims to create general-purpose embeddings applicable to a multitude of downstream tasks. These representations have become integral to numerous applications, including text-to-image generation (e.g., Stable Diffusion [2]), Large Multimodal Models [3, 4, 5], and open-set object detection [6, 7]. The widespread adoption and diverse applications of learned representations like CLIP emphasize the need to understand their underlying semantics. Therefore, effective explanation methods are required to characterize the information encoded within these representations. As these learned embeddings are inherently task-agnostic and depend only on the input data, their explanations should therefore describe the information they hold, regardless of any specific downstream application.

Explaining general-purpose representations presents a significant challenge for traditional eXplainable AI (XAI) techniques. Methods such as GradCAM [8], LIME [9], and Integrated Gradients [10] are fundamentally designed to explain model predictions for specific tasks or classes. For instance, Grad-CAM utilizes class-specific gradients to generate activation maps, while LIME approximates local decision boundaries. Critically, this inherent focus on task-specific predictions prevents their generalization to explaining the underlying representations. A more recent line of work [11, 12] approaches

---

[*]Corresponding author. Email: mythai@cise.ufl.edu

39th Conference on Neural Information Processing Systems (NeurIPS 2025).

this challenge using matrix-factorization techniques, such as Non-negative Matrix Factorization (NMF), to find low-rank decompositions of embeddings. While such methods can produce bases that effectively reconstruct the original embeddings, the resulting basis is not inherently interpretable and requires further analysis to discern its meaning.

To address these limitations, we propose EXPLAIN-R (EXPLAIN-Representations), a novel method designed to generate interpretable conceptual explanations for CLIP image representations, independent of downstream tasks. By design, EXPLAIN-R discovers human-understandable concepts from each input image. Our method utilizes these discovered concepts to construct a surrogate representation, which is then trained to mimic the behavior of the original CLIP embeddings. Post-training, our method computes concept scores that quantify the influence of each discovered concept on an input's representation. Our contributions are as follows:

- We introduce EXPLAIN-R, a novel method for learning a faithful surrogate representation of CLIP image embeddings, which is formed by linearly combining interpretable concepts, and propose a theoretically-motivated algorithm for its training.

- Extensive quantitative experiments across multiple datasets validate the faithfulness of EXPLAIN-R, demonstrating that the surrogate representation accurately preserves the predictive behavior of the original CLIP model.

- We establish via a user study that EXPLAIN-R produces explanations considered relevant to the input image, sufficiently complete to explain the model's capabilities, and useful for overall model comprehension.

## 2 Related Work

**Interpreting CLIP vision encoder.** Existing methodologies for explaining CLIP models can be broadly categorized into several distinct lines of work, each adopting a unique approach. Pixel attribution techniques [13, 14, 15, 16] identify input regions influencing model outputs via heatmaps. Being visually accessible, they can explain the "where" but fall short in explaining the "what" [11]. For instance, a visual explanation highlighting the poodle in an image does not clarify if the model recognizes a "poodle" specifically or merely a "dog". Mechanistic interpretability methods [17, 18, 19] pursue a different strategy, associating human concepts with internal model components (e.g., neurons, attention heads). Concept Bottleneck Model [20, 21, 22, 23] aims to substitute the opaque representation with an explicit concept layer. However, CBMs typically focus on task-specific predictions rather than explaining the versatile representation that is applicable to multiple tasks. In contrast, our method focuses on generating interpretable concept-based explanations for these abstract image representations directly. Furthermore, our method can be extended to provide concept explanations for zero-shot tasks without additional training.

**Concept-based representation explanations.** One specific line of work aims to find a concept basis spanning the representation space using matrix decomposition [12, 11, 24, 25]. These methods typically factorize dataset embeddings via SVD or NMF into a smaller basis and corresponding coefficient matrices. Although capable of achieving low reconstruction error, the resulting basis vectors are not inherently interpretable by design. Their semantic meaning needs to be inferred through subsequent analysis, like visualizing top activating inputs [12, 11]. SpLiCE [26], the most relevant prior work, decomposes CLIP image features into sums of text features using the CLIP text encoder. However, imperfections in the text encoder can lead to spurious or unrelated concepts (as observed in Section 4.2), potentially causing confusion for the user. Our method distinguishes itself from these approaches through its design for inherent interpretability: the discovered concepts are, by construction, explicitly linked to the input image, and do not rely on the text encoder. Furthermore, our proposed similarity-matching training strategy yields superior faithfulness while preserving the surrogate model's simplicity and interpretability.

## 3 Methodology

This section details EXPLAIN-R, a novel method for interpreting CLIP image representations through a learned concept surrogate model. EXPLAIN-R constructs a surrogate embedding from a linear combination of interpretable concepts grounded in the image. The process encompasses three key

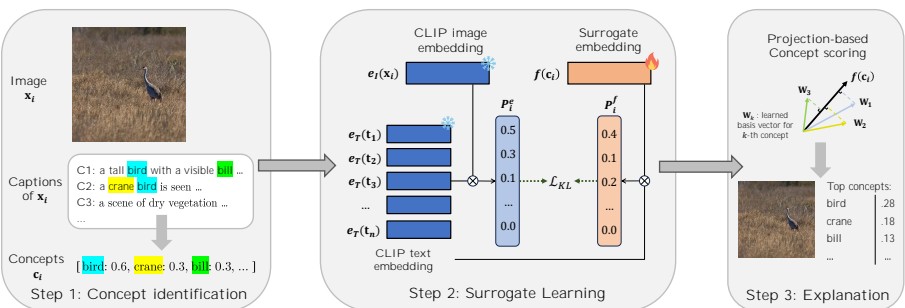

Figure 1: EXPLAIN-R methodology overview. (1) **Concept Identification**: Image captions are processed into a concept vector $\mathbf{c}_i$. (2) **Surrogate Learning**: The surrogate $f$, with parameters $\mathbf{W},\mathbf{b}$, is trained to mimic the CLIP image embedding by matching image-to-text ($P_i^f, P_i^e$) and text-to-image (not visualized) similarity distributions using a KL divergence loss. Symbol $\bigotimes$ denotes cosine similarity followed by softmax. (3) **Explanation**: Explanatory concept scores for the representation of the image $\mathbf{x}_i$ are obtained by projecting the rows of $\mathbf{W}$ onto $f(\mathbf{c}_i)$ and weighing by $\mathbf{c}_i$.

steps, visualized in Fig. 1: (1) discovery of interpretable concepts from the image dataset to build a vocabulary $C$; (2) learning a surrogate model that transforms these image-specific concepts into an embedding that maintains similarity with text embeddings; and (3) computation of concept scores to quantify each concept's contribution, thus providing an explanation for the image's CLIP representation.

**Notations.** Let $\{(\mathbf{x}_i, \mathbf{t}_i)\}_{i=1}^n$ be an image-text dataset. CLIP's image encoder $e_I(\cdot)$ and text encoder $e_T(\cdot)$ yield $L_2$-normalized representations in $\mathbb{R}^d$. For each image $\mathbf{x}_i$, we obtain the concept vector $\mathbf{c}_i \in \mathbb{R}^{|C|}$ and form the augmented dataset $\mathcal{S} = \{(\mathbf{x}_i, \mathbf{c}_i, \mathbf{t}_i)\}$. The surrogate model $f : \mathbb{R}^{|C|} \to \mathbb{R}^d$ is defined as $f(\mathbf{c_i}; \mathbf{W}, \mathbf{b}) = \sigma(\mathbf{c}_i\mathbf{W} + \mathbf{b})$. Here $\mathbf{W} \in \mathbb{R}^{|C| \times d}$ is a matrix where rows are the (learned) concept basis vectors, $\mathbf{b} \in \mathbb{R}^d$ is a bias, and $\sigma(\cdot)$ is the $L_2$ normalization operator. The surrogate function linearly transforms the concepts into the embedding space, which is then normalized.

### 3.1 Captions-based concept identification

Explaining generic, task-agnostic image representations with open-ended concepts necessitates concept identification methods that are scalable, task-independent, and yield intuitive, human-centric concepts. Our work is directed towards developing such an approach.

However, existing concept identification paradigms are largely incompatible with achieving these combined objectives. For instance, (i) supervised concept detectors [27, 28, 29] require labeled data for predefined concepts, limiting their scalability for open-ended concept discovery. (ii) LLM-based concept generation [30, 22, 21] is typically designed to yield task-specific concepts (e.g., for class discrimination), which lacks the task-independence required for general representation understanding. (iii) Furthermore, directly employing CLIP's alignment scores [31, 23] can produce unintuitive concept associations, as CLIP's training objective aligns images with full sentences or captions rather than prioritizing the descriptive accuracy of individual words or concepts in a human-like manner.

To obtain explanations that are intuitive, broadly applicable, and thus more aligned with our objectives, we advocate sourcing concepts from data reflecting typical human image descriptions. Image captions are ideal as they tend to describe a diverse range of image features (such as objects, attributes, and actions). This descriptive characteristic means that concepts derived from captions are largely purpose-neutral, making them well-suited for explaining task-agnostic representations. This caption-based approach also offers benefits such as transparency into how concept scores are derived. While human-annotated captions are preferred, those from advanced models [3, 4] (trained to emulate human styles via metrics correlated with human judgment [32, 33]) offer a scalable alternative. Based on these ideas, we formulate the concept representation for an image $\mathbf{x}_i$ as a vector $\mathbf{c}_i \in [0, 1]^{|C|}$, where each component $\mathbf{c}_{i,k}$ quantifies the prominence of the $k$-th concept from a global vocabulary C within human-like descriptions of $\mathbf{x}_i$, empirically measuring its descriptive likelihood.

The construction of the vocabulary $C$ and the concept vectors $\mathbf{c}_i$ proceeds systematically. First, we compile an initial, comprehensive vocabulary by extracting all nouns, verbs, and adjectives from the image captions across the dataset. This raw vocabulary is then pruned to create C by removing overly frequent terms, as these are often less discriminative, guided by an adjustable threshold to control the desired concept sparsity (i.e., the average number of active concepts per image). Subsequently, for each image $\mathbf{x}_i$, its concept vector component $\mathbf{c}_{i,k}$ is computed as $z/m$, where $m$ is the number of captions associated with $\mathbf{x}_i$, and $z$ is the frequency of the $k$-th concept (from $C$) within those $m$ captions. The resulting concept vector, $\mathbf{c}_i$, reflect the qualities of an image that are emphasized in typical human descriptions.

## 3.2  Learning interpretable surrogate representation

Previous works [26, 11] on finding concept-based explanations focus on high-fidelity numerical reconstruction of the embeddings. For example, the baseline SpLiCE attempts to maximize the cosine similarity between the reconstructed and the original embedding. While a numerically identical reconstruction would entail perfect faithfulness, achieving this ideal is often infeasible in the presence of interpretability constraints like sparsity. Moreover, given that CLIP is trained to capture bimodal relationships, unimodal similarities can behave unintuitively, as pointed out in [34]. This makes optimizing for the similarity between an original embedding and its reconstruction a less reliable method to capture CLIP's bimodal behavior.

Our proposed approach alternatively focuses on what is learned with CLIP's contrastive objective. We posit that CLIP's contrastive training, which requires distinguishing an image's paired caption from numerous alternatives, leads to a rich similarity distribution over texts, and vice versa. This pattern of similarity to other inputs, we argue, offers insights into CLIP's behavior that previous methods overlooked. Consequently, our surrogate model, $f$, is trained to reproduce this distribution of similarities, without specific downstream task supervision.

We now formally state the surrogate learning problem. The distribution of image-to-text similarities of the $i$-th image for the original model is defined as:

$$P_i^e(j) = \frac{\exp(e_I(\mathbf{x}_i)^\top e_T(\mathbf{t}_j)/\tau)}{\sum_k \exp(e_I(\mathbf{x}_i)^\top e_T(\mathbf{t}_k)/\tau)}. \tag{1}$$

where $\tau$ is the temperature. The similarity distribution for the surrogate is analogously given as:

$$P_i^f(j) = \frac{\exp(f(\mathbf{c}_i)^\top e_T(\mathbf{t}_j)/\tau)}{\sum_k \exp(f(\mathbf{c}_i)^\top e_T(\mathbf{t}_k)/\tau)}. \tag{2}$$

Likewise, the text-to-image similarity distribution $Q_j^e$ and $Q_j^f$ are obtained by modifying the normalizing factor in the denominator of (1) and (2) to sum over image indices. Faithfulness in a surrogate model necessitates that its output similarity distributions, $P_i^f$ and $Q_j^f$, closely mirror those of the target model, $P_i^e$ and $Q_j^e$, as these distributions determine predicted probabilities and outputs. To promote such faithfulness, we propose minimizing the Kullback-Leibler (KL) divergence between these corresponding distribution pairs. This makes the distributions more similar by reducing the information available to distinguish between them [35]. In this case, the KL divergence also has an intuitive interpretation: the term $D_{KL}(P_i^e \parallel P_i^f)$, for instance, is the average difference of log probability between $P_i^e$ and $P_i^f$, weighted by $P_i^e$. Taking the expectation of the divergence over the dataset, we obtain the final loss function:

$$\mathcal{L}(\mathbf{W}, \mathbf{b}) = \frac{1}{2n}\left(\sum_{i=1}^n D_{KL}(P_i^e \parallel P_i^f) + \sum_{j=1}^n D_{KL}(Q_j^e \parallel Q_j^f)\right). \tag{3}$$

We note that this loss function have been used in conjunction with various other mechanisms (weak/strong augmentation, momentum encoders, etc.) for self-supervised learning from scratch [36]. However, our motivation (encouraging faithfulness) and use case (XAI) is completely different, and to the best our ability, the idea of training a surrogate by distilling instance-instance similarity has not

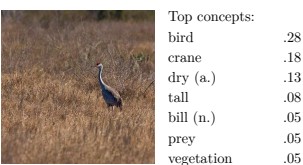
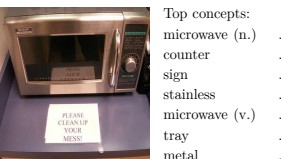
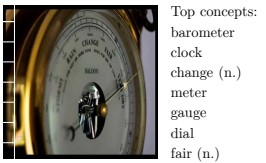

| Top concepts: | | Top concepts: | | Top concepts: | |
|---|---|---|---|---|---|
| bird | .28 | microwave (n.) | .40 | barometer | .15 |
| crane | .18 | counter | .08 | clock | .13 |
| dry (a.) | .13 | sign | .08 | change (n.) | .11 |
| tall | .08 | stainless | .06 | meter | .08 |
| bill (n.) | .05 | microwave (v.) | .05 | gauge | .06 |
| prey | .05 | tray | .04 | dial | .06 |
| vegetation | .05 | metal | .04 | fair (n.) | .05 |

Figure 2: Example images from ImageNet and their top concepts, ranked by CLIP representation concept score $v_i$. In case of ambiguity, the concepts are annotated with their part of speech. The images are labeled as *crane*, *microwave*, and *barometer* respectively. We visualize the top seven concepts for brevity; our surrogate representations contain 15-20 concepts.

been explored for XAI. Finally, note that our choice of a linear surrogate promotes interpretability by allowing computation of projection-based concept scores as in Section 3.4. It is also consistent with assumptions in existing works [12, 11], including the baseline [26].

### 3.3 Algorithm for Surrogate Learning

In this section, we present our algorithm for optimizing (3). As we will show, naively using mini-batch optimization for (3) leads to a biased gradient. This is a common problem for contrastive losses and their variants, which existing works circumvent by either using a larger batch size (for example, OpenAI uses a batch size of 32,768 on 256 GPUs to train the CLIP model [1]), using a queue of past samples [37, 36], or using non-contrastive losses [38, 39]. Instead, our solution to correct this bias does not require large batch size and it is theoretically motivated.

We will focus only on $D_{KL}(P_i^e \parallel P_i^f)$ since the procedure for optimizing $D_{KL}(Q_j^e \parallel Q_j^f)$ can be derived analogously. From the definition of KL divergence, the first part of (3) can be equivalently written as (c.f. Appendix B.2 for more detailed derivation):

$$\frac{1}{|\mathcal{S}|^2} \sum_{i \in \mathcal{S}} \sum_{j \in \mathcal{S}} \frac{1}{g(e_I, e_T; \mathbf{x}_i, \mathbf{t}_j, \mathcal{S})} \cdot \log g(f, e_T; \mathbf{c}_i, \mathbf{t}_j, \mathcal{S}), \tag{4}$$

where $g(f, e_T; \mathbf{c}_i, \mathbf{t}_j, \mathcal{S}) = \frac{1}{|\mathcal{S}|} \sum_{k \in \mathcal{S}} \exp\left(\frac{f(\mathbf{c}_i)^\top e_T(\mathbf{t}_k) - f(\mathbf{c}_i)^\top e_T(\mathbf{t}_j)}{\tau}\right)$. The gradient w.r.t. $f$ is then given by:

$$\frac{1}{|\mathcal{S}|^2} \sum_{i \in \mathcal{S}} \sum_{j \in \mathcal{S}} \frac{1}{g(e_I, e_T; \mathbf{x}_i, \mathbf{t}_j, \mathcal{S})} \cdot \frac{1}{g(f, e_T; \mathbf{c}_i, \mathbf{t}_j, \mathcal{S})} \cdot \nabla g(f, e_T; \mathbf{c}_i, \mathbf{t}_j, \mathcal{S}).$$

At each iteration, we only have access to a mini-batch of triplets $\mathcal{B} = \{(\mathbf{x}_i, \mathbf{c}_i, \mathbf{t}_i)\}$ of batch size $B$. The obtained mini-batch gradient estimator is simply replacing $\mathcal{S}$ with $\mathcal{B}$ in the above equation. However, due to the non-linearity of the reciprocal function $1/\cdot$, the expectation over $\mathcal{B}$ is not equal to the true gradient. Thus the mini-batch estimator is a biased estimator of the true gradient. To solve this problem, we use two moving average sequences $u$ and $v$ to approximate $g(e_I, e_T; \mathbf{x}_i, \mathbf{t}_j, \mathcal{S})$ and $g(f, e_T; \mathbf{c}_i, \mathbf{t}_j, \mathcal{S})$ respectively:

$$u_{t+1,i} = (1 - \gamma_1)u_{t,i} + \gamma_1 \frac{1}{|\mathcal{B}_t|} \sum_{k \in \mathcal{B}_t} \exp\left(\frac{e_I(\mathbf{x}_i)^\top e_T(\mathbf{t}_k)}{\tau}\right), \tag{5}$$

$$v_{t+1,i} = (1 - \gamma_2)v_{t,i} + \gamma_2 \frac{1}{|\mathcal{B}_t|} \sum_{k \in \mathcal{B}_t} \exp\left(\frac{f(\mathbf{x}_i)^\top e_T(\mathbf{t}_k)}{\tau}\right), \tag{6}$$

where $\gamma_1, \gamma_2 \in (0, 1]$ are two hyperparameters. Then we can approximate $g(e_I, e_T; \mathbf{x}_i, \mathbf{t}_j, \mathcal{S})$ and $g(f, e_T; \mathbf{c}_i, \mathbf{t}_j, \mathcal{S})$ using $u_{t+1,i}/\exp\left(e_I(\mathbf{x}_i)^\top e_T(\mathbf{t}_j)/\tau\right)$ and $v_{t+1,i}/\exp\left(f(\mathbf{c}_i)^\top e_T(\mathbf{t}_j)/\tau\right)$, respectively. The benefit of using the moving average technique is that now we can guarantee that the distance between the estimators and the true values ($g(e_I, e_T; \mathbf{x}_i, \mathbf{t}_j, \mathcal{S})$ and $g(f, e_T; \mathbf{c}_i, \mathbf{t}_j, \mathcal{S})$) diminishes to 0 in expectation, instead of remaining at a constant level $\mathcal{O}(1/B)$ if the moving average technique is not used [40, 41]. We present the pseudocode and full derivation in Appendix B.2.

| Concepts for 'mixing bowl' | |
|---|---|
| clay | .09 |
| bowl | .05 |
| pottery | .04 |
| wheel | .02 |
| potter | .02 |
| spin | .01 |
| hand | .01 |

(a) Image of class "potter's wheel" misclassified as "mixing bowl" by CLIP. Note the significant contribution of the *bowl* concept, which is a main predictor for the "mixing bowl" class, as seen in Fig. 6.

| Concepts for 'barometer' | | Concepts for 'wall clock' | |
|---|---|---|---|
| barometer | .060 | clock | .054 |
| clock | .047 | barometer | .042 |
| meter | .025 | meter | .020 |
| gauge | .022 | change (n.) | .019 |
| dial | .020 | gauge | .016 |
| change (n.) | .016 | dial | .016 |
| mechanical | .013 | mechanical | .012 |

(b) Class concept scores of the same image for different classes. The contribution of concepts varies in strength depending on the target class.

Figure 3: Class concept attribution score for ImageNet images.

## 3.4 Interpreting representations with projection-based concept scores

This section introduces a method for attributing the CLIP image representations to their underlying concepts. Standard attribution methods, such as SHAP [42], typically rely on the notion of a prediction to operate. In contrast, our approach leverages the linearity of the explainer $f$ to efficiently computes concept attribution scores $v_i(k)$. Building upon this, we further develop a cross-modal attribution score $v_{i,j}(k)$ to quantify how concept $k$ modulates the similarity between image $i$ and text $j$.

We begin by expanding the functional form of the surrogate $f(\mathbf{c}_i) = \sigma(\mathbf{W}\mathbf{c}_i + \mathbf{b})$, which yields:

$$f(\mathbf{c}_i) = \sigma\left(\sum_{k \in C} \mathbf{c}_{i,k}\mathbf{W}_k + \mathbf{b}\right), \tag{7}$$

where $\mathbf{W}_k$ is the column of $\mathbf{W}$ corresponding to the embedding of the $k$-th concept. Equation (7) shows that each concept contributes a term $\mathbf{c}_{i,k}\mathbf{W}_k$ to the pre-normalized representation. A term is considered to have a greater influence if it aligns with the direction of the final representation $f(\mathbf{c}_i)$. We quantify this alignment with the scalar projection of $\mathbf{c}_{i,k}\mathbf{W}_k$ onto $f(\mathbf{c}_i)$, which we define as the initial score: $\tilde{v}_i(k) = \mathbf{c}_{i,k}\mathbf{W}_k^\top f(\mathbf{c}_i)$. Note that summing the projection recovers the pre-normalization magnitude, i.e., $\sum_{k \in C} \tilde{v}_i(k) + \mathbf{b}^\top f(\mathbf{c}_i) = \|\mathbf{W}\mathbf{c}_i + \mathbf{b}\|$. To obtain scores reflecting the relative contribution of each concept while ignoring the bias term, we normalize them by $\lambda_i = \left(\|\mathbf{W}\mathbf{c}_i + \mathbf{b}\| - \mathbf{b}^\top f(\mathbf{c}_i)\right)^{-1}$. We define the final attribution score $v_i(k)$ as:

$$v_i(k) := \lambda_i \tilde{v}_i(k) = \lambda_i \mathbf{c}_{i,k}\mathbf{W}_k^\top f(\mathbf{c}_i). \tag{8}$$

This score $v_i(k)$ quantifies the normalized contribution of the $k$-th concept, based on the linear relationship between the $k$-th concept embedding and the final surrogate representation $f(\mathbf{c}_i)$, such that $\sum_{k \in C} v_i(k) = 1$.

**Cross-model concept scores.** We adapt this projection-based approach to attribute the cross-modal similarity between the $i$-th image and the $j$-th text to individual concepts $k$. Recall that the text embedding is $e_T(\mathbf{t}_j)$, and the similarity is computed as $s_{i,j} = f(\mathbf{c}_i)^\top e_T(\mathbf{t}_j)$. We aim to understand how each concept $k$ influences the similarity $s_{i,j}$. Analogous to before, we project the contribution vector onto the direction of the text embedding $e_T(\mathbf{t}_j)$ to obtain the unnormalized score $\tilde{v}_{i,j}(k) = \mathbf{c}_{i,k}\mathbf{W}_k^\top e_T(\mathbf{t}_j)$. The final cross-modal attribution score is then:

$$v_{i,j}(k) = \lambda_{i,j}\mathbf{c}_{i,k}\mathbf{W}_k^\top e_T(\mathbf{t}_j), \tag{9}$$

in which the normalization factor $\lambda_{i,j} = \frac{f(\mathbf{c}_i)^\top e_T(\mathbf{t}_j)}{(W\mathbf{c}_i)^\top e_T(\mathbf{t}_j)}$ makes the scores $v_{i,j}(k)$ sum to $s_{i,j}$ over all concepts.

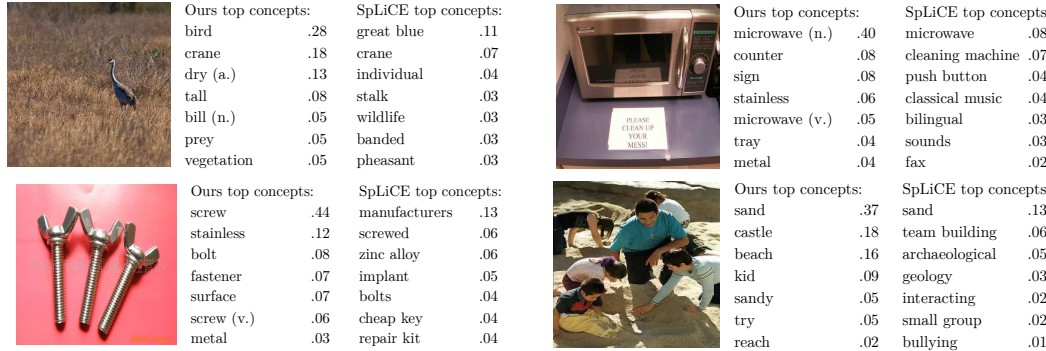

| Ours top concepts: | | SpLiCE top concepts: | |
|---|---|---|---|
| bird | .28 | great blue | .11 |
| crane | .18 | crane | .07 |
| dry (a.) | .13 | individual | .04 |
| tall | .08 | stalk | .03 |
| bill (n.) | .05 | wildlife | .03 |
| prey | .05 | banded | .03 |
| vegetation | .05 | pheasant | .03 |

| Ours top concepts: | | SpLiCE top concepts: | |
|---|---|---|---|
| microwave (n.) | .40 | microwave | .08 |
| counter | .08 | cleaning machine | .07 |
| sign | .08 | push button | .04 |
| stainless | .06 | classical music | .04 |
| microwave (v.) | .05 | bilingual | .03 |
| tray | .04 | sounds | .03 |
| metal | .04 | fax | .02 |

| Ours top concepts: | | SpLiCE top concepts: | |
|---|---|---|---|
| screw | .44 | manufacturers | .13 |
| stainless | .12 | screwed | .06 |
| bolt | .08 | zinc alloy | .06 |
| fastener | .07 | implant | .05 |
| surface | .07 | bolts | .04 |
| screw (v.) | .06 | cheap key | .04 |
| metal | .03 | repair kit | .04 |

| Ours top concepts: | | SpLiCE top concepts: | |
|---|---|---|---|
| sand | .37 | sand | .13 |
| castle | .18 | team building | .06 |
| beach | .16 | archaeological | .05 |
| kid | .09 | geology | .03 |
| sandy | .05 | interacting | .02 |
| try | .05 | small group | .02 |
| reach | .02 | bullying | .01 |

Figure 4: Comparison between our concepts and the baseline. Images are from the SUN397 (lower right) and ImageNet (remaining) datasets. The top-left image depicts a crane (bird), while great blue and pheasant are other species of birds. We show only the top seven concepts for each image. Concept scores from our method sum to one for each image; SpLiCE's scores do not have this property.

## 4 Experiments

We conduct experiments using the CLIP ViT-B/32 [1] model to demonstrate that explanations generated by our method are faithful, interpretable, and useful for users.

**Datasets.** We use the COCO 2017 [43] validation set, Flickr30k [44], SUN397 [45] test set, and ImageNet validation set [46] to study the CLIP model's behavior. These datasets cover a variety of image themes, including objects in context, internet images, scene understanding, and object recognition. Details and experiments on more datasets can be found in the Appendix.

**Setup.** For datasets that contain captions (COCO and Flickr30k), we use the captions as the text $t_i$. For image-only datasets, we use the BLIP-2 [3] OPT 2.7B model fine-tuned on COCO to obtain 10 captions per image. Then, we perform concept identification (Section 3.1) to obtain 15-20 concepts per image. We train the surrogate with Algorithm 1 for 150 epochs, with batch size 1024, default temperature $\tau = 0.1$, and $\gamma_1 = \gamma_2 = 0.9$. The optimizer used is AdamW with learning rate $10^{-3}$. All experiments are performed on a single A100 GPU. For the baseline SpLiCE [26], we use the official implementation linked in the paper. We obtain a similar sparsity (average number of concepts per image) to our method by setting the $l_1$ penalty for each dataset to facilitate a fair comparison. The vocabulary used for SpLiCE is based on LAION and has a size of 15,000.

### 4.1 Explanation faithfulness

Emphasizing our focus on post-hoc explanation, we assess the surrogate's faithfulness quantitatively by comparing the predictions made using the surrogate representation against those of the original image embedding. Faithfulness is measured using conventional metrics (e.g., accuracy), with the target model's prediction treated as the ground truth. We focus on zero-shot tasks as opposed to linear probing, since the former relies directly on the image representations and does not depend on a particular trained probe's behavior. For zero-shot classification, we report the accuracy. For zero-shot retrieval tasks, we report the rsum metrics [47, 48] (defined as R@1 + R@5 + R@10) as a concise summary, instead of individual Recall@K (R@K) values. Detailed retrieval metrics and performance metrics on the original tasks are provided in Appendix A.1 and A.2.

Table 1: Faithfulness of our method and the baseline on zero-shot retrieval and zero-shot classification. Higher values are better. The means and $2\sigma$ intervals are computed over five runs. We note that the official implementation of SpLiCE is deterministic.

| Method | COCO val | Flickr30k | SUN397 | ImageNet val |
|---|---|---|---|---|
| SpLiCE [26] | 393.18 | 381.41 | 59.78 | 51.33 |
| Ours | **482.94** $\pm$ 4.08 | **476.47** $\pm$ 3.21 | **62.77** $\pm$ 0.22 | **53.38** $\pm$ 0.21 |

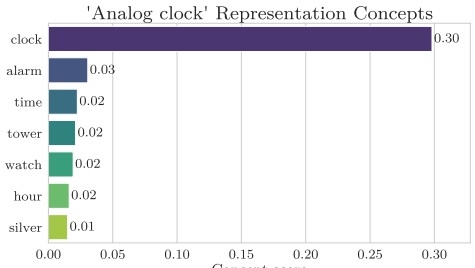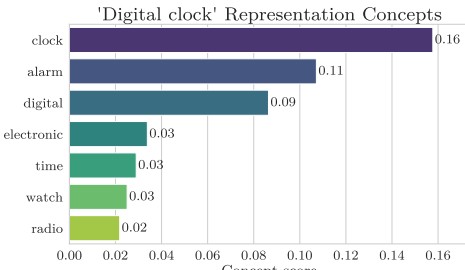

Figure 5: Average representation concept scores for images in ImageNet "Analog clock" and "Digital clock" classes.

As shown in Table 1, our surrogate model consistently outperforms the baseline in faithfulness across diverse datasets and tasks, indicating its effectiveness in replicating the target model's behavior. This success demonstrates that faithful surrogate concept representations can be achieved without solely relying on the minimization of reconstruction mean-squared error, a strategy prevalent in the baseline and other methods [11, 20, 12].

## 4.2 Qualitative assessment of representation concepts

We visualize the representation concepts generated by our methods and the baseline to evaluate their relevance to the image content. We show several images from ImageNet and SUN397 and their corresponding explanations in Fig. 4, with more details and examples in Appendix A.3. Our method predominantly discovers concepts highly pertinent to the image content. Our explanations suggest that the CLIP image representation attends to both the primary subject and its surrounding context; the top-ranked concept often identifies the main object with high score, while others capture secondary objects or specific attributes of the primary subject. Conversely, the baseline (SpLiCE) sometimes produces concepts that, while potentially related, are not depicted (e.g., "great blue," "cleaning machine," "geology"), or are entirely unrelated (e.g., "classical music," "cheap key").

**Representation Concept score Histograms.** Per-image concept scores (exemplified in Fig. 4) can be aggregated to offer insights into how the CLIP model represents classes of images in a dataset. To illustrate, Figure 5 presents the aggregated top seven concept scores for two related classes: "analog clock" and "digital clock". The analysis reveals distinct patterns: "analog clock" image embeddings are predominantly characterized by the "clock" concept, whereas "digital clock" images show a more uniform concept distribution. Notably, while several top concepts like "alarm", "time", and "watch" are common to both, the concepts "digital" and "electronic" effectively distinguish the two classes.

## 4.3 Class prediction analysis

This section illustrates how class concept scores ($v_{i,j}$) can be applied to understand model predictions for a specific class, enabling misprediction analysis. To reiterate for clarity in this context, these scores, $v_{i,j}$, quantify how each concept from an image $i$ contributes to its similarity $s_{i,j}$ with class $j$, and are defined such that their sum equals $s_{i,j}$. This class-specific nature contrasts with task-agnostic

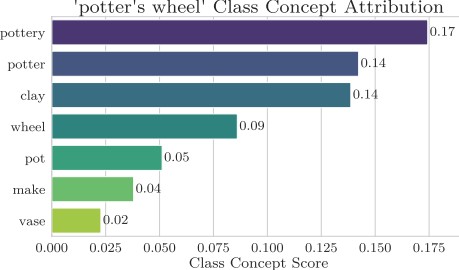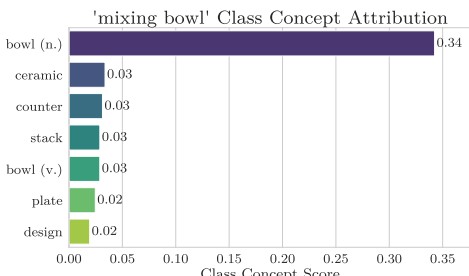

Figure 6: Average class concept scores for ImageNet classes "potter's wheel" and "mixing bowl".

representation concept scores ($v_i$), which describe the overall image content. Figure 3b analyzes the class concept scores ($v_{i,j}$) for a specific image; the task-agnostic representation concepts ($v_i$) for this same image are depicted in the rightmost panel of Figure 2. For this image, the "barometer" class prediction is primarily driven by the "barometer" concept (rank 1) followed by "clock" (rank 2), while the "wall clock" class prediction prioritizes "clock" (rank 1) over "barometer" (rank 2). This shows that the contribution of individual concepts to $v_{i,j}$ varies significantly with the class.

**Misprediction analysis.** Analysis of class concept scores for mispredicted images can reveal concepts responsible for incorrect classifications. For instance, Figure 3a presents an ImageNet image of class "potter's wheel" that both CLIP and the surrogate misclassified as "mixing bowl" (a bowl used for cooking). The concepts contributing to this prediction are shown alongside the image. To determine the typical concepts used for each class, we aggregated the class concept scores $v_{i,j}$ for both "potter's wheel" and "mixing bowl", visualizing the results in Figure 6. This combined information suggests that the "bowl" concept is the primary driver of the misclassification. Specifically, the aggregation indicates that "bowl" is, on average, the dominant concept for predicting "mixing bowl", while it is notably absent from the top concepts associated with "potter's wheel".

### 4.4 User study

We performed a small scale user study to evaluate the interpretability of our explanations (results in Fig. 7), largely following the protocol of [26] but with modified criteria. Users were shown 20 random ImageNet images and the top ten concepts from EXPLAIN-R and SpLiCE. The evaluation centered on three criteria: (1) **Relevance**: the degree to which concepts pertain to the input image; (2) **Completeness**: the extent to which explanations reflect the semantic richness and task-agnostic characteristics of the image representations; and (3) **Utility**: the perceived usefulness of each method for understanding the model's behavior. Responses were captured using a 5-point Likert scale. User study findings reveal a significant preference for our explanations across all evaluated criteria. The statistical significance of these results was established via a one-sided t-test, employing a significance level of $p < 0.05$. Additional details of the user study protocol are available in the Appendix.

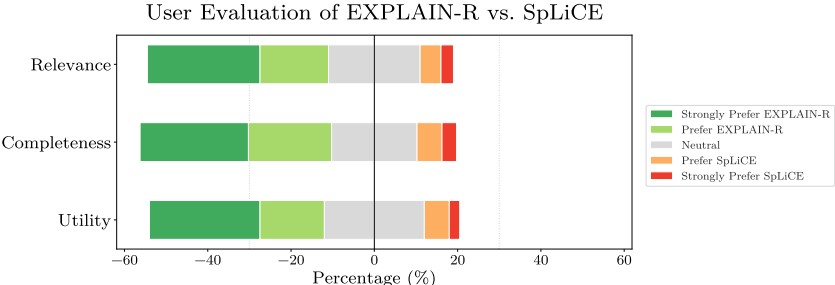

Figure 7: User study results comparing representation explanations generated by EXPLAIN-R and SpLiCE. Overall, users indicated a clear preference for EXPLAIN-R's explanations across all three evaluated criteria.

## 5 Discussion

In this work, we show that the instance similarities learned by CLIP can be utilized to generate concept explanations for CLIP image embeddings. Quantitative experiments demonstrated EXPLAIN-R's superior faithfulness over baselines in preserving original model predictions, while a user study confirmed its explanations are more relevant, complete, and useful for model understanding. These results suggest that our similarity-matching approach offers a promising direction for developing more faithful and human-aligned explanations for general-purpose representations.

**Limitations** Our approach, like prior studies [26, 11, 12], assumes that concepts interact linearly in the embedding space. While EXPLAIN-R significantly improves faithfulness over the baseline, perfect fidelity remains challenging. This gap may stem from several factors: the target model might learn concepts that are not easily captured by concise textual descriptions, or there are some

non-linear concept interactions. A further consideration is our use of a captioning model instead of human captions. While this makes the approach more feasible and scalable, it also risks the captioner hallucinating concepts not present in the image, which can negatively affect the relevance of the explanations. Selecting a properly evaluated captioning model that is suitable for the target image domain can help minimize this issue.

## Acknowledgments and Disclosure of Funding

We express appreciation for the volunteers in our user study. We thank the anonymous reviewers for their constructive feedback. This work is partially supported by National Science Foundation (NSF) grants SCH-2123809 and SCH-2306572, Learning Engineering Virtual Institute grant G-23-2137070 and University of Florida Presidential Strategic Funding Award. Any opinions, findings, and conclusions or recommendations expressed in this paper, however, are those of the authors and do not necessarily reflect the views of the funding agency.

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

## Appendix Contents

# A  Additional results

## A.1  Retrieval faithfulness metrics

Table 2: COCO retrieval faithfulness metrics. For our method, we report the mean over 5 runs.

| Method | COCO I→T | | | COCO T→I | | |
|---|---|---|---|---|---|---|
| | R@1 | R@5 | R@10 | R@1 | R@5 | R@10 |
| SpLiCE [26] | 38.12 | 72.40 | 86.00 | 38.26 | 72.74 | 85.66 |
| Ours | **61.99** | **89.31** | **95.51** | **55.03** | **86.64** | **94.46** |

Table 3: Flickr30k retrieval faithfulness metrics. For our method, we report the mean over 5 runs.

| Method | Flickr30k I→T | | | Flickr30kT→I | | |
|---|---|---|---|---|---|---|
| | R@1 | R@5 | R@10 | R@1 | R@5 | R@10 |
| SpLiCE [26] | 40.25 | 70.41 | 80.65 | 40.50 | 69.56 | 80.04 |
| Ours | **66.37** | **86.32** | **91.26** | **60.31** | **82.79** | **89.42** |

Table 2 and Table 3 shows the detailed R@K metrics for COOC and Flickr30k of our method and SpLiCE, which is supplementary to Table 1 of the main paper.

## A.2  Task performance metrics

Table 4: Performance metrics. For our method, we report the mean over 5 runs.

| Method | COCO I→T | | | COCO T→I | | | SUN397 | ImageNet |
|---|---|---|---|---|---|---|---|---|
| | R@1 | R@5 | R@10 | R@1 | R@5 | R@10 | | |
| SpLiCE [26] | 32.26 | 62.94 | 76.44 | 29.92 | 59.22 | 73.00 | 52.41 | 42.91 |
| CLIP ViT-B/32 | 51.90 | 81.26 | 90.24 | 47.48 | 77.10 | 87.08 | **60.71** | **61.91** |
| Ours | **58.03** | **85.68** | **93.19** | **56.30** | **84.98** | **93.09** | 57.98 | 52.37 |

Table 4, 5 presents a comparison of performance metrics (evaluated against dataset ground truth) for our method, SpLiCE, and the original CLIP model. The baseline is setup similar to the experiment in Tab. 1. As can be seen in Table 4, our method consistently outperforms the baseline SpLiCE across tasks given the same sparsity.

Furthermore, although designed for post-hoc explanation, our surrogate representation exhibits the ability to sometimes outperform the CLIP model it explains on the zero-shot tasks we tested, despite never having access to the dataset labels and only being trained on the similarities produced by the CLIP model. We hypothesize that performance improvement (when they exists) stems from the increased robustness of concept-based inputs, which may be less susceptible to common image degradations such as occlusion, blurriness, or general noise, compared to raw image inputs.

## A.3  More visualizations

In Figure 8, we provide a more comprehensive list of concepts of the images visualized in Fig. 4.

## A.4  More datasets

In Table 6, we provide more zero-shot classification results on more datasets (Flowers102, Food101). Our method continues to yield consistent improvements on these fine-grained classification tasks. In Figure 9, we visualize some results from these datasets.

Table 5: Performance metrics (continued). For our method, we report the mean over 5 runs.

| Method | Flickr30k I→T | | | Flickr30k T→I | | |
|---|---|---|---|---|---|---|
| | R@1 | R@5 | R@10 | R@1 | R@5 | R@10 |
| SpLiCE [26] | 36.43 | 65.03 | 75.36 | 36.83 | 63.50 | 74.20 |
| CLIP ViT-B/32 | 67.17 | 88.90 | 93.80 | 63.60 | 86.78 | 92.26 |
| Ours | **74.53** | **92.58** | **96.24** | **73.60** | **92.67** | **96.30** |

Table 6: Faithfulness metrics for more datasets.

| Method | Flowers102 | Food101 |
|---|---|---|
| SpLiCE | 26.12 | 51.80 |
| Ours | **37.73** | **63.62** |

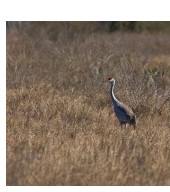

Ours top concepts:
bird .28
crane .18
dry (a.) .13
tall .08
bill (n.) .05
prey .05
vegetation .05
walk (v.) .04
feather .03
morning .03
faced .03
standing .02
grey .01
pointed .01
middle .00

SpLiCE top concepts:
great blue .11
crane .07
individual .04
stalk .03
wildlife .03
banded .03
pheasant .03
foraging .02
foreground .01
hunting .01
identifying .01
marsh .01
wetlands .01
south africa .01
woodpecker .01

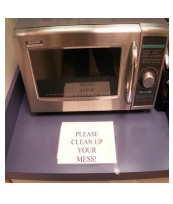

Ours top concepts:
microwave (n.) .40
counter .08
sign .08
stainless .06
microwave (v.) .05
tray .04
metal .04
clean (a.) .03
mess (n.) .03
instruction .03
say .02
state (v.) .02
room .02
floor .02
steel .01

SpLiCE top concepts:
microwave .08
cleaning machine .07
push button .04
classical music .04
bilingual .03
sounds .03
fax .02
culinary .02
blank sign .02
cd player .02
depending .02
claimed .02
sanitary .02
laundry room .02
non slip .02

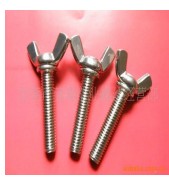

Ours top concepts:
screw .44
stainless .12
bolt .08
fastener .07
surface .06
screw (v.) .06
metal .03
wrench .03
pink .03
washer .03
use .02
steel .02
shape .02
animal .01
sitting -0.01

SpLiCE top concepts:
manufacturers .13
screwed .06
zinc alloy .06
implant .05
bolts .04
cheap key .04
repair kit .04
screw .03
jual .03
stud earrings .03
bullets .02
capitals .02
wrench .01
clips .01
nickel .01

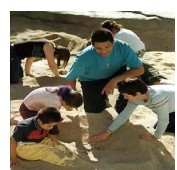

Ours top concepts:
sand .37
castle .18
beach .16
kid .09
sandy .05
try .05
reach .02
family .02
hand .02
adult .01

SpLiCE top concepts:
sand .13
team building .06
archaeological .05
geology .03
interacting .02
small group .02
bullying .01
children play .01
little children .13
bath shower .00

Figure 8: More comprehensive depiction of concept list from our method and the baseline.

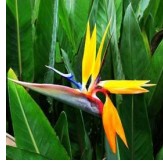

Ours top concepts:
paradise .45
bird .26
colorful .08
yellow .07
tropical .06
blue .04
color .01
garden .01
surround .01
lush .01

SpLiCE top concepts:
beautiful flower .08
tropical leaves .06
tropical fish .05
silk flower .04
costa rica .03
cute birds .02
form .02
exotic .01
biodiversity .01
beautiful nature .01

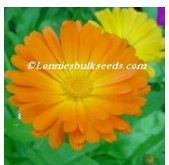

Ours top concepts:
orange (n.) .64
yellow (v.) .11
orange (a.) .08
purple (n.) .07
daisy .07
lush .01
blooming .01
garden .01
surround .01
beautiful .00

SpLiCE top concepts:
beautiful flower .08
orange flower .06
jual .05
orange yellow .03
macro .03
seed oil .02
online flower .02
online store .02
nature background .02
anne .01

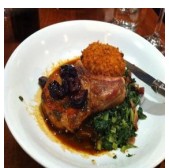

Ours top concepts:
steak .38
green .15
entrée .14
broccoli .10
cook .09
knife .06
wine .05
service .05
many -0.02
type -0.01

SpLiCE top concepts:
lamb .10
appetizer .06
Italian food .06
pork .04
grouse .03
meal .03
sizzling .02
yummy .02
side dish .01
portion .01

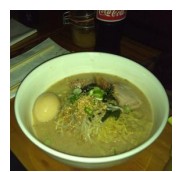

Ours top concepts:
soup .37
noodle .33
pork .09
coca .06
cola .06
soda .04
restaurant .02
bottle .02
coke .01
wood .01

SpLiCE top concepts:
ramen .15
second round .10
comfort food .06
noodle soup .03
Japanese food .03
bomb .02
delicious .01
Korean .01
broth .01
order .00

Figure 9: Visualizations for samples from the Flowers102 dataset (upper) and Food101 dataset (lower).

# B Method details

## B.1 Concept identification details

We redescribe our concept identification process step-by-step with more details:

1. (If not already present) obtain the captions for each image by using a captioning model

2. Extract nouns, verbs, and adjectives concepts via part-of-speech tagging with the `nltk` library

3. Filter out words not presenting in WordNet, infrequent and overly frequent concepts to obtain the vocabulary $C$

4. Estimate concept prevalence score for each image-concept pair to obtain the concept vectors $\mathbf{c}_i$

## B.2 Algorithm for Surrogate Learning

Below is the pseudocode for the algorithm described in Section 3.3.

---
**Algorithm 1:** Algorithm for Surrogate Learning

---
**Input:** CLIP encoders $e_I, e_T$, surrogate model $f$, dataset $\mathcal{S}$, concepts $\mathcal{C}$, temperature $\tau$, batch size $B$, number of iterations $T$

1 **for** $t = 0, \ldots, T-1$ **do**
2     Sample a mini-batch of triplets $\mathcal{B}_t = \{(\mathbf{x}_i, \mathbf{c}_i, \mathbf{t}_i)\}$ from the dataset
3     Update $u_{t+1,i}, v_{t+1,i}$ using (5) and (6) for $i \in \mathcal{B}_t$
4     Set $u_{t+1,i} = u_{t,i}, v_{t+1,i} = v_{t,i}$ for $i \notin \mathcal{B}_t$
5     Compute gradient estimator w.r.t. $f_t$:

$$\frac{1}{|\mathcal{B}_t|^2} \sum_{i \in \mathcal{B}_t} \sum_{j \in \mathcal{B}_t} \frac{\exp\left(\frac{e_I(\mathbf{x}_i)^\top e_T(\mathbf{t}_j)}{\tau}\right)}{u_{t+1,i}} \cdot \frac{\exp\left(\frac{f(\mathbf{c}_i)^\top e_T(\mathbf{t}_j)}{\tau}\right)}{v_{t+1,i}} \cdot \nabla g(f_t, e_T; \mathbf{c}_i, \mathbf{t}_j, \mathcal{B}_t).$$

6     Update $f_{t+1}$ from $f_t$ using an optimizer

---

We now present the full derivation of Algorithm 1. We will focus only on $D_{KL}(P_i^e \parallel P_i^f)$ since the procedure for optimizing $D_{KL}(Q_j^e \parallel Q_j^f)$ can be derived analogously. From the definition of KL divergence, we can write the first part of (3) as

$$\frac{1}{2n} \sum_{i=1}^{n} D_{KL}(P_i^e \parallel P_i^f) = -\frac{1}{2n} \sum_{i=1}^{n} \sum_{j=1}^{n} P_i^e(j) \log P_i^f(j) + \frac{1}{2n} \sum_{i=1}^{n} \sum_{j=1}^{n} P_i^e(j) \log P_i^e(j).$$

Note that $e$ is fixed and we want to optimize only $f$, we will discard the second term on the right hand side since it does not involve $f$. Plugging (1) and (2) into the above equation, we get

$$
\begin{aligned}
&-\frac{1}{2n} \sum_{i=1}^{n} \sum_{j=1}^{n} P_i^e(j) \log P_i^f(j) \\
&= -\frac{1}{2n} \sum_{i=1}^{n} \sum_{j=1}^{n} \frac{\exp(e_I(\mathbf{x}_i)^\top e_T(\mathbf{t}_j)/\tau)}{\sum_k \exp(e_I(\mathbf{x}_i)^\top e_T(\mathbf{t}_k)/\tau)} \cdot \log \frac{\exp(f(\mathbf{c}_i)^\top e_T(\mathbf{t}_j)/\tau)}{\sum_k \exp(f(\mathbf{c}_i)^\top e_T(\mathbf{t}_k)/\tau)} \\
&= \frac{1}{2n} \sum_{i=1}^{n} \sum_{j=1}^{n} \left( \sum_{k=1}^{n} \exp\left(\frac{e_I(\mathbf{x}_i)^\top e_T(\mathbf{t}_k) - e_I(\mathbf{x}_i)^\top e_T(\mathbf{t}_j)}{\tau}\right) \right)^{-1} \\
&\qquad\qquad \cdot \log \sum_{k=1}^{n} \exp\left(\frac{f(\mathbf{c}_i)^\top e_T(\mathbf{t}_k) - f(\mathbf{c}_i)^\top e_T(\mathbf{t}_j)}{\tau}\right).
\end{aligned}
\tag{10}
$$

Recall that $\mathcal{S}$ denotes the whole dataset, and

$$g(f, e_T; \mathbf{c}_i, \mathbf{t}_j, \mathcal{S}) := \frac{1}{n}\sum_{k=1}^{n} \exp\left(\frac{f(\mathbf{c}_i)^{\top}e_T(\mathbf{t}_k) - f(\mathbf{c}_i)^{\top}e_T(\mathbf{t}_j)}{\tau}\right)$$

$$= \frac{1}{n}\sum_{k=1}^{n} \exp\left(\frac{f(\mathbf{c}_i)^{\top}e_T(\mathbf{t}_k)}{\tau}\right) \bigg/ \exp\left(\frac{f(\mathbf{c}_i)^{\top}e_T(\mathbf{t}_j)}{\tau}\right).$$

Then (10) can be equivalently written as

$$\frac{1}{|\mathcal{S}|^2}\sum_{i\in\mathcal{S}}\sum_{j\in\mathcal{S}} \frac{1}{g(e_I, e_T; \mathbf{x}_i, \mathbf{t}_j, \mathcal{S})} \cdot \log g(f, e_T; \mathbf{c}_i, \mathbf{t}_j, \mathcal{S}). \tag{11}$$

The gradient w.r.t. $f$ is given by

$$\frac{1}{|\mathcal{S}|^2}\sum_{i\in\mathcal{S}}\sum_{j\in\mathcal{S}} \frac{1}{g(e_I, e_T; \mathbf{x}_i, \mathbf{t}_j, \mathcal{S})} \cdot \frac{1}{g(f, e_T; \mathbf{c}_i, \mathbf{t}_j, \mathcal{S})} \cdot \nabla g(f, e_T; \mathbf{c}_i, \mathbf{t}_j, \mathcal{S}).$$

Since we only have access to one mini-batch of data $\mathcal{B}$ at each iteration, the obtained mini-batch gradient estimator is

$$\frac{1}{|\mathcal{B}|^2}\sum_{i\in\mathcal{B}}\sum_{j\in\mathcal{B}} \frac{1}{g(e_I, e_T; \mathbf{x}_i, \mathbf{t}_j, \mathcal{B})} \cdot \frac{1}{g(f, e_T; \mathbf{c}_i, \mathbf{t}_j, \mathcal{B})} \cdot \nabla g(f, e_T; \mathbf{c}_i, \mathbf{t}_j, \mathcal{B}).$$

However, due to the non-linearity of the reciprocal function, the expectation over $\mathcal{B}$ is not equal to the true gradient. Thus the mini-batch estimator is biased. To address this problem, we use two moving average sequences $u$ and $v$ to approximate $g(e_I, e_T; \mathbf{x}_i, \mathbf{t}_j, \mathcal{S})$ and $g(f, e_T; \mathbf{c}_i, \mathbf{t}_j, \mathcal{S})$ respectively:

$$u_{t+1,i} = (1 - \gamma_1)u_{t,i} + \gamma_1 \frac{1}{|\mathcal{B}_t|} \sum_{k\in\mathcal{B}_t} \exp\left(\frac{e_I(\mathbf{x}_i)^{\top}e_T(\mathbf{t}_k)}{\tau}\right),$$

$$v_{t+1,i} = (1 - \gamma_2)v_{t,i} + \gamma_2 \frac{1}{|\mathcal{B}_t|} \sum_{k\in\mathcal{B}_t} \exp\left(\frac{f(\mathbf{x}_i)^{\top}e_T(\mathbf{t}_k)}{\tau}\right),$$

where $\gamma_1, \gamma_2 \in (0, 1]$ are two hyperparameters. Then we can approximate $g(e_I, e_T; \mathbf{x}_i, \mathbf{t}_j, \mathcal{S})$ and $g(f, e_T; \mathbf{c}_i, \mathbf{t}_j, \mathcal{S})$ using

$$u_{t+1,i} \bigg/ \exp\left(\frac{e_I(\mathbf{x}_i)^{\top}e_T(\mathbf{t}_j)}{\tau}\right), \quad \text{and } v_{t+1,i} \bigg/ \exp\left(\frac{f(\mathbf{c}_i)^{\top}e_T(\mathbf{t}_j)}{\tau}\right).$$

### B.3   Experimental details

**Datasets.**   For the Flickr30k dataset, we explain on the full dataset. For the SUN397 dataset, we use the first official testing split. For the COCO 2017 and ImageNet dataset, we use the validation split. Following [26], for computational efficiency, the retrieval metrics are computed in batches of size 1000 and averaged over the full dataset.

**Training.**   We use the Amsgrad variant of the AdamW optimizer with learning rate $10^{-3}$ and weight decay $10^{-6}$. During training, we distill image-text similarities within the batch following Algorithm 1. We perform augmentations on both modalities: selecting a random caption as text augmentation, and random center crop and horizontal flip as image augmentation.

### B.4   User study details

The user study involved 10 volunteers who did not receive monetary compensation. The interface is shown in Figure 10. Notably, we observed that SpLiCE's weights are often lower than that of us, since they do not sum to one. To reduce the chance of the user differentiating the two methods based on the weights, we scaled SpLiCE's weights for each individual image so that their sum equals ours.

The study was ruled exempt by our institution's IRB, as no more than minimal risk is posed to the participants. No identifying information was collected.

| Criteria | CI | p-value |
|---|---|---|
| Relevance | $1.61 \pm 0.54$ | $9 \times 10^{-8}$ |
| Completeness | $1.69 \pm 0.72$ | $1 \times 10^{-6}$ |
| Utility | $1.66 \pm 0.62$ | $4 \times 10^{-7}$ |

Table 7: The p-values and confidence intervals (CI) for hypothesis testing in our user study. A value of 1 denotes strong preference for EXPLAIN-R, averaged across the 20 shown samples. The hypothesis tested is whether the population mean is less than 3, which denotes neurality (no preference for SpLiCE or EXPLAIN-R).

## B.5 Broader impacts

Our work addresses the problem of interpreting CLIP image representations in a task independent manner. EXPLAIN-R provides a tool for users and researchers to inspect the semantic content of the representation and provide a simple, intuitive summarization of the learned concepts for each image. For the users, this will enhance transparency and trustworthiness in multimodal representations, which traditionally relies on downstream evaluation. Researchers can use EXPLAIN-R to inspect individual embeddings and model predictions, as well as aggregate them over the dataset to obtain a more holistic view of the concepts learned by the model.

Figure 10: The interface for our user study. For each input image, we show the top 10 concepts from both methods, along with the weights. We scale SpLiCE's weights so they have the same mean as ours.

