# OpenReview forum: "Advancing Interpretability of CLIP Representations with Concept Surrogate Model"
_NeurIPS.cc/2025/Conference — NeurIPS 2025 poster_

### Official Review · Reviewer_3CxX · 2025-06-23

**Clarity:** 3
**Significance:** 2
**Originality:** 2
**Rating:** 4
**Confidence:** 4

**Summary:**

This paper introduces EXPLAIN-R, a novel method for generating concept-based explanations for general-purpose CLIP image representations. The core idea is to train a simple, interpretable surrogate model to mimic CLIP's behavior. Instead of traditional reconstruction loss, the surrogate is trained to match the instance-instance similarity distributions of the original CLIP model, measured by KL divergence. While the paper presents an interesting new learning objective and shows promising results in a user study, I recommend rejection due to its insufficient experimental validation, strong underlying assumptions, and failure to acknowledge closely related prior work.

**Questions:**

see Strengths And Weaknesses

**Ethical Concerns:**

["NO or VERY MINOR ethics concerns only"]

**Final Justification:**

See my discussion notes.

**Limitations:**

see Strengths And Weaknesses

**Quality:**

2

**Strengths And Weaknesses:**

Strengths

The paper addresses the important and challenging problem of explaining task-agnostic representations. The core idea of training a surrogate model by matching similarity distributions, rather than by direct embedding reconstruction, is novel and theoretically well-motivated. It aligns more closely with the contrastive nature of CLIP itself.

Weaknesses

Despite these strengths, the paper has several major weaknesses that prevent me from recommending acceptance.

1. Insufficient experimental validation: This is the most significant flaw. The empirical evidence provided is not comprehensive enough to fully support the claims of a new, general-purpose explanation method:

a. Limited datasets: The experiments are confined to general-domain datasets (COCO, ImageNet, etc.). The method's effectiveness is not tested on more challenging fine-grained recognition tasks (e.g., CUB-200 for birds, Oxford Flowers), where concept-based explanations are arguably most needed and most difficult to generate. The proposed concept discovery method, which relies on statistical extraction from general captions, is unlikely to capture the subtle, domain-specific concepts required for these tasks.

b. Single baseline comparison: The quantitative evaluation relies on a comparison against a single baseline, SpLiCE. To robustly demonstrate superiority, a new method should be compared against a wider range of relevant approaches, such as prototype-based models (e.g., ProtoPNet) or concept bottleneck models (CBMs), even if they require adaptation.

c. Lack of ablation studies: The method's performance is heavily dependent on the quality of its concept vocabulary, which in turn depends on an external captioning model (BLIP-2) and heuristic filtering. The paper lacks a rigorous ablation study on how the quality of captions (e.g., human-authored vs. machine-generated) or the choice of filtering heuristics impacts the final explanation quality.

d. Failure to cite and differentiate from highly related work: The paper fails to acknowledge or discuss highly relevant and concurrent work, which significantly undermines its claim to originality.
The work Align2Concept (ACM MM 2024)[1] shares a remarkably similar goal: providing language-guided, concept-level explanations for visual recognition by bridging visual and textual modalities through CLIP's multimodal space. Both methods aim to align visual parts with textual concepts to create interpretable models.
While the technical approaches differ (surrogate learning vs. manifold alignment with prototypes), the high-level motivation and conceptual framework are nearly identical. The omission of such a closely related work is a major oversight and prevents a fair assessment of this paper's true contribution to the field.

e. Unjustified strong assumption of linearity: The entire framework is built on the strong assumption that concepts in CLIP's embedding space interact linearly. This is a significant simplification of a complex, high-dimensional space where non-linear interactions are highly probable. The paper acknowledges this as a limitation but fails to discuss its potential impact, which could lead to misleading or overly simplistic explanations that do not capture the true reasoning of the model.

[1] Align2Concept: Language Guided Interpretable Image Recognition by Visual Prototype and Textual Concept Alignment, ACM MM2024

---

> ### Author Rebuttal · Authors · 2025-07-31
>
> We thank you for your comments. Below we address your concerns:
>
> **Q1: Limited evaluation datasets and lack of “challenging” datasets**
>
> A: First, we clarify that we evaluated and reported results on fine-grained recognition tasks. These results are in Appendix A.4 of the supplementary material. They include the Flowers102 (Oxford flowers) and Food101 datasets. In both datasets we outperform the baseline. We included visualizations which show images and their concept in Fig. 9 in the supp. matt.
>
> We emphasize that “popular” datasets are not necessarily “easy”. In the main paper, we report results on 4 datasets (COCO, Flickr30k, SUN397, ImageNet). The SUN397 dataset contains many similar scene categories (e.g., abbey, basilica, cathedral, church) and can be considered fine-grained ([1], [2], [3]). The ImageNet also contains multiple breeds of the same animal (e.g., 120 dog breeds [4]), which also makes it fine-grained [5].
>
> **Q2: Single baseline comparison and not citing Align2Concept**
>
> A: We would like to make a key distinction here: our work positions itself as an explainer, not a replacement to the CLIP model with better performance/interpretability. Our goal is to understand the CLIP image encoder, not to replace it. It is due to this positioning that we value faithfulness more than task performance throughout our paper.
>
> Many related works fall in the category of creating inherently interpretable methods that aim to replace opaque methods. While they might share similarities to us (concept extraction, concept alignment, …) the ultimate goals are completely different. For example, a replacement model is good if it performs better on downstream tasks, or it is more interpretable than the base CLIP model. It does not need to have the same predictions as the base model (i.e., it does not need to be faithful). A surrogate explainer is only good if it is faithful to the base model, which gives insight into how the base model actually works. The two different philosophies renders comparisons pointless.
>
> Furthermore, our work is motivated by the need to explain general-purpose CLIP representation (discussed in answer to Q2 of reviewer hVGW). Traditional task-specific methods such as CBMs cannot be simply “adapted” to work with this setting.
>
> The above reasons explain why we compare our method with the most similar baseline SpLiCE, which also priorities explaining over replacement.
>
> Regarding the work Align2Concept, its abstract states: “Our experiments show that the ProCoNet provides higher accuracy and better interpretability compared to the single-modality interpretable model.” This confirms that this method belongs to the category of inherently interpretable methods, as discussed earlier. While it is suitable to discuss this work in our paper, the differing motivation and does not impact our paper’s novelty.
>
> The other suggested “relevant approach” ProtoPNet also has the same issue: it is a whole new architecture that is interpretable, and it is not an explainer for existing models.
>
> **Q3: Linear representation assumption/hypothesis**
>
> A: We note that there is evidence for this hypothesis in the literature, including the baseline SpLiCE and other works ([6], [7], [8]).
>
> During the development of the paper, we also experimented with a non-linear surrogate. This resulted in greatly increased training time, without significant improvement in faithfulness.
>
> The literature evidence coupled with our developmental result suggest that the hypothesis is not entirely unreasonable. While it might be a simplification, it allowed us to gain insights into the representation. Studying the exact geometry of hidden representation is an active research direction and is out of scope of our work.
>
> **Q4: Concept extraction ablation study**
>
> A: Our proposed concept extraction scheme is simple, computationally efficient, and effective. In the same spirit as our replies to Q1 of reviewer iH42 and Q2 of reviewer hVGW, we acknowledge the concept extraction can be improved, but it is not the key/innovative part of our method.
>
> In Section 3.1 of the paper, we stated that human captions are ideal since they contain the concepts that humans are likely to use when describing the image and using an image captioning models train on human data achieves our goal. Even with BLIP2-generated concepts, our method remains effective, demonstrated via downstream faithfulness (Table 1 of main paper) and user preference (Figure 7 of main paper).
>
> [1] M. Zanella and I. B. Ayed, "On the Test-Time Zero-Shot Generalization of Vision-Language Models: Do we Really need Prompt Learning?," 2024 IEEE/CVF Conference on Computer Vision and Pattern Recognition (CVPR), Seattle, WA, USA, 2024, pp. 23783-23793.
>
> [2] Maniparambil, Mayug, et al. "Enhancing clip with gpt-4: Harnessing visual descriptions as prompts." Proceedings of the IEEE/CVF international conference on computer vision. 2023.
>
> [3] Y. Liang, Y. Pan, H. Lai, W. Liu and J. Yin, "Deep Listwise Triplet Hashing for Fine-Grained Image Retrieval," in IEEE Transactions on Image Processing, vol. 31, pp. 949-961, 2022.
>
> [4] Russakovsky, O., Deng, J., Su, H. et al. ImageNet Large Scale Visual Recognition Challenge. Int J Comput Vis 115, 211–252 (2015).
>
> [5] X. -S. Wei et al., "Fine-Grained Image Analysis With Deep Learning: A Survey," in IEEE Transactions on Pattern Analysis and Machine Intelligence, vol. 44, no. 12, pp. 8927-8948.
>
> [6] Trager, M., Perera, P., Zancato, L., Achille, A., Bhatia, P., & Soatto, S. (2023, October). Linear spaces of meanings: Compositional structures in vision-language models. In Proceedings of the IEEE/CVF International Conference on Computer Vision (ICCV) (pp. 15395–15404).
>
> [7] Fel, T., Picard, A., Béthune, L., Boissin, T., Vigouroux, D., Colin, J., Cadène, R., & Serre, T. (2023, June). CRAFT: Concept recursive activation factorization for explainability. In Proceedings of the IEEE/CVF Conference on Computer Vision and Pattern Recognition (CVPR) (pp. 2711–2721).
>
> [8] Papadimitriou, Isabel & Su, Huangyuan & Fel, Thomas & Saphra, Naomi & Kakade, Sham & Gil, Stephanie. (2025). Interpreting the Linear Structure of Vision-language Model Embedding Spaces. 10.48550/arXiv.2504.11695.

---

### Official Review · Reviewer_hVGW · 2025-06-24

**Clarity:** 4
**Significance:** 3
**Originality:** 3
**Rating:** 5
**Confidence:** 3

**Summary:**

The authors propose EXPLAIN-R, a new method to tackle interpretability when using CLIP representations, independantly of the downstream taskl. The method consists in extracting concepts from captions, generated by a captioning model, that are used as input of a surrogate model that aims at reproducing the distribution of image-text similarities. They conducted extensive experiments on several datasets.

**Questions:**

1. Could you please clarify how different captions were generated using a single captioning model? Specifically, was the goal of generating 10 captions to eliminate noise, or perhaps to capture a wider range of concepts? Additionally, were these captions distinct from one another in terms of content and phrasing?

2. Did you consider using multiple models to generate diverse captions?

3. Could you please give more motivation behind the task independence?

**Ethical Concerns:**

["NO or VERY MINOR ethics concerns only"]

**Final Justification:**

Based on the authors’ responses to my questions and the assessments provided by the other reviewers, I have decided to maintain my original rating.

**Limitations:**

Yes

**Quality:**

4

**Strengths And Weaknesses:**

**Strengths:** The method, EXPLAIN-R, introduced in this work to tackle the interpretability of CLIP representations, which is a key challenge given the widespread use of the CLIP model. The authors provide a clear and intuitive explanation of the methodology, making it easy to follow. Regarding the experiment section, the inclusion of a user study strengthens the experimental evaluation.

**Weaknesses:** As the authors themselves acknowledge, relying on captions generated by the same captioning model for concept extraction can be a limitation (see Question 2). Furthermore, I am uncertain about how different captions were produced using a single model and whether these captions were truly distinct from one another (see Question 1). Additionally, I remain unconvinced about the necessity of task-independent interpretability. I would think that the relevance of explanatory concepts likely depends on the specific task. For instance, distinguishing birds from planes may require different concepts than differentiating between two bird species (see Question 3).

---

> ### Author Rebuttal · Authors · 2025-07-31
>
> We thank you for your comments and support! Below we address your concerns:
>
> **Q1: Details and goals of generating multiple captions**
>
> A: Our captioning model is a VLM (vision-language model), so it works very similar to an LLM. Specifically, we increase the temperature to encourage diversity and use top-p sampling to generate different captions given the same input.
>
> To illustrate our point, we show the first 5 captions generated for an ImageNet image:
>
> "a photo of an older gentleman holding a large fish in his hands",
>
>  "a photo of a man in khaki pants is holding a big fish",
>
>  "a photo of a man posing with a big fish that he caught",
>
>  "a photo of the man is about to give away a huge fish he caught",
>
>  "a photo of man holds large fish next to trees and stream",
>
> The different captions both introduced unique concepts (old man, khaki, trees, …) and reinforce concepts that certainly exists (big fish, the action of the man holding the fish, …)
>
> The goal of using 10 captions is to capture more accurately the distribution of concepts that humans would use to describe the image. In that sense, it both reduces noise and encourages rarer concepts.
>
> **Q2: On using multiple models to generate captions**
>
> A: This is a practical idea that should give even more diverse captions. However, due to batching, using 1 model to generate 10 captions will be much quicker than, for example, using 10 models to generate 1 caption each. We experimented and found that our procedure, described in the answer to the previous question, hits a good balance between efficiency and diversity. We selected the BLIP2 model due to its good performance in captioning ([1]).
>
> **Q3: Motivation behind task independence**
>
> A: The CLIP image encoder generates a general-purpose embedding for each image that can be used for various downstream tasks. We draw parallel to humans. When given an image, a human can memorize certain features about that image that they can use to distinguish from other images. If asked to perform a certain task (e.g., find out which species of bird), then the human will pay more attention to certain features (e.g., fur color, size, environment, etc.).
>
> We posit that the CLIP image encoder works in the same way, i.e., it encodes a set of concepts independent of task in the representation. Then, for each task, a certain subset of concepts will be more important than others. We realize this idea in the paper. We define the notion of the contribution of a concept to an embedding in Eq. (8) in the paper, and this score is what you see in Fig. 2 and 4. This approach allows converting embeddings into a interpretable semantic basis, which enable analysis that task-specific methods cannot. In Fig. 5 we show an example of this kind of analysis: we find that the notion of “analog clock” is mostly dependent on the notion of “clock” itself, while the notion of “digital clock” involves the coexistence of the notion of “clock” and other notions such as “alarm” or “digital”.
>
> Finally, we provide a mathematically meaningful way to convert embedding contribution to class-dependent contribution in Eq. (9), which aligns with the intuition in the first paragraph.
>
> [1] Sarto, S.; Cornia, M.; and Cucchiara, R. 2025. Image Captioning Evaluation in the Age of Multimodal LLMs: Challenges and Future Perspectives. In Proceedings of the International Joint Conference on Artificial Intelligence (IJCAI-25).

---

> > ### Comment · Reviewer_hVGW · 2025-08-06
> >
> > Thank you very much for the clarification. I maintain my original grade.

---

> > > ### Author Response · Authors · 2025-08-08
> > >
> > > Dear Reviewer hVGW,
> > >
> > > We hope the clarification was informative for you and useful. Please feel free to let us know if you think the paper/presentation could be changed/improved in any way.
> > >
> > > Best,
> > > The authors

---

### Official Review · Reviewer_iH42 · 2025-07-03

**Clarity:** 2
**Significance:** 3
**Originality:** 2
**Rating:** 4
**Confidence:** 4

**Summary:**

In this work, the authors propose EXPLAIN-R, an interpretability for CLIP that attempts to faithfully reconstruct the behavior of
the vision encoder while operating instead over concepts. In doing so, they provide a concept-based explanation of the vision
encoder while also remaining faithful to the downstream performance of the CLIP model.

**Questions:**

* Do you think image captions contain all the relevant concepts in an image? For example, in the image example in Fig. 1,
your captions cover “A tall bird with a visible bill” and “dry vegetation”. However, I could imagine a relevant concept being “sunset” or “sunny weather” despite annotators generally describing the foreground and background over other visually-represented concepts such as the time of day or the weather.
 * I would be curious to also compare performance with the ground truth labels instead of CLIP — you optimize for faithfulness so it is expected that you do well on this metric.
 * It appears to me that this paper is more focused on explaining the behavior of CLIP’s image encoder over concepts than the concept extraction from images itself — your method for concept extraction is only a small section in the paper. I think this might be important to explain in the framing of your work.

**Ethical Concerns:**

["NO or VERY MINOR ethics concerns only"]

**Limitations:**

Yes

**Paper Formatting Concerns:**

* Bullet two of contributions, you should say “We [verb] extensive quantitative contributions”
 * Minor spelling and grammar concerns, I would definitely give the draft a re-read, but here are a couple that stood out to
me:
  * Line 62: Concept Bottleneck Model aims to → Concept Bottleneck Models aim to
  * Line 190: efficiently computes → efficiently compute

**Quality:**

3

**Strengths And Weaknesses:**

## Strengths

 * Training a model to be faithful to the vision encoder is a novel idea and helps develop off of previous concept-based
explanations of CLIP . The ability to get cross-modal concept scores and use them in class-specific concept decompositions for
misprediction analysis was interesting as well.
 * The experiments validate the faithfulness of the method and the qualitative examples also seem to validate the method.

## Weaknesses
 * Your description of how to compute concept vectors in 3.1 is very clear but I would appreciate an equation, figure, or
algorithm to diagram out the computation of each concept. In figure 1 the specific computation of the weights of the
concept vector are abstracted away .
 * To some extent this is a concept bottleneck model - you extract concepts and then train a model on these concepts for a
task (in this case, replicating CLIP’s image embedding). You may want to address this connection in more detail.

---

> ### Author Rebuttal · Authors · 2025-07-31
>
> We thank you for your encouraging comments! We will address the formatting issues in the final version. Below we address your concerns:
>
> **Q1: Do you think image captions contain all relevant concepts in an image?**
>
> A: Extending our reasoning in the third paragraph of Section 3.1, when we use captions, we want them to be a) as similar to human captions as possible, and b) varied enough to capture human description variety.
>
> To achieve a), we believe that using models trained on existing annotated data is the best. To achieve b), we sample multiple captions for variety. To answer your specific inquiry, we believe that a more sophisticated sampling strategy that penalizes repetitions should lead to more varied concepts, including more non-visual ones. In general, concept extraction is not the key part of our work, but better concept extraction would definitely benefit the explanation.
>
> **Q2: On performance with ground truth labels**
>
> A: We reported the task performance (i.e., metrics computed with the dataset ground truth instead of CLIP outputs) in the supplementary material Appendix A.2. We report results for 3 datasets: COCO, SUN397, and ImageNet. In some cases, our surrogate outperforms the original CLIP model. In all cases, our surrogate is more performant than the baseline.
>
> **Q3: The focus of the paper**
>
> A: Indeed, our method is focused on explaining the behavior of the CLIP image encoder. Our idea is that each image has a set of concepts that humans care about. This set is independent of the model used. Each model would then “weigh” this set differently depending on how they are trained. This explains our framework of model-agnostic concept extraction, followed by learning the weight for each concept.
>
> **Q4: On computation of concept vectors**
>
> A: We provided details on how to compute the coefficients for each concept in the last paragraph of Section 3.1 of the main paper and Appendix B.1 in the supp. matt. We will rewrite it as an algorithm in the final version to make it clearer and more concise.

---

> > ### Comment · Reviewer_iH42 · 2025-08-04
> > **Reviewer Response**
> >
> > Thank you for answering my questions! As long as you frame the paper to focus on interpreting the vision encoder I'm on board.

---

> > > ### Author Response · Authors · 2025-08-08
> > >
> > > Dear Reviewer iH42,
> > >
> > > We are glad that we were able to make the positioning of our paper clearer to you! Please let us know if there are any more questions.
> > >
> > > Best,
> > > The authors

---

### Decision · Program_Chairs · 2025-09-17

**Decision:**

Accept (poster)

**Comment:**

The paper deals with the problem of interpretability of Contrastive Language-Image Pre-training models (CLIP models), which is addressed by using a surrogate representation of concepts present in an image.

The paper was considered by three reviewers. The final ratings were: BA - A - BA

The reviewers liked the idea of training a model to assess the vision encoded and highlight the good experimental evaluation. Most issues were addressed in the rebuttal. The AC follows the consensus of the reviewer voting and recommends accepting the paper. The AC would encourage the authors to integrate the findings of the rebuttal in the CR version of the paper.